# TaskForce: Cooperative Multi-agent Reinforcement Learning for Multi-task Optimization

## Abstract

Multi-task learning (MTL) involves the simultaneous optimization of multiple task-specific losses, often leading to gradient conflicts and scale imbalances that result in negative transfer. While existing multi-task optimization methods attempt to mitigate these challenges, they either lack the stochasticity needed to escape poor local minima or fail to explicitly address conflicts at the gradient level. In this work, we propose TaskForce, a novel multi-task optimization framework incorporating cooperative multi-agent reinforcement learning (MARL), where agents learn to find an effective joint optimization strategy based on their respective task gradients and losses. To keep the optimization process compact yet informative, agents observe a summary of the training dynamics that consists of the gradient Gram matrix—capturing both gradient magnitudes and pairwise alignments—and task loss values. Each agent then predicts the balancing parameters that determine the weight of their contribution to the final gradient update. Crucially, we design a hybrid reward function that incorporates both gradient-based signals and loss improvement dynamics, enabling agents to effectively resolve gradient conflicts and avoid poor convergence by considering both direct gradient information and the resulting impact on loss reduction. TaskForce achieves consistent improvements over state-of-the-art MTL baselines on NYU-v2, Cityscapes, and QM9, demonstrating the promise of cooperative MARL in complex multi-task scenarios.

## 1 Introduction

Multi-task learning (MTL) (Caruana, 1997) is a paradigm in machine learning where a single model is trained to solve multiple tasks simultaneously. By sharing representations across tasks, MTL encourages knowledge transfer and leverages commonalities between related tasks. This shared representation can lead to improved generalization (Zhang & Yang, 2021), particularly when some tasks suffer from limited labeled data. Furthermore, MTL has the potential to reduce computational cost and memory footprint by consolidating multiple models into a unified architecture. Building on these advantages, MTL has shown promise in enhancing both performance and robustness across various benchmarks, such as several vision tasks (Ye & Xu, 2022a;b; Choi et al., 2024) and natural language processing (Hashimoto et al., 2016; McCann et al., 2018).

Despite its advantages, MTL often causes negative transfer (Crawshaw, 2020)–an ill-posed problem that arises when jointly learning unrelated or weakly correlated tasks, leading to one task impairing the learning of others. A major contributor to negative transfer is gradient conflict (Yu et al., 2020; Wang et al., 2020; Liu et al., 2021a), where the gradient directions derived from different task losses point in opposing or diverging directions in parameter space. This can lead to unstable updates or biased convergence toward tasks with dominant gradients (Navon et al., 2022), limiting the effectiveness of MTL frameworks deployed in real-world systems.

To alleviate these challenges, previous studies have explored various strategies, including architectural modifications that adjust the sharing ratio of parameters (Misra et al., 2016; Sun et al., 2020; Choi & Im, 2023) and task grouping strategies that cluster related tasks (Zamir et al., 2018; Fifty et al., 2021). Among these, multi-task optimization (MTO) methods have shown strong performance by effectively addressing core issues such as gradient conflicts and scale dominance, which are ma-

jor causes of negative transfer. These MTO methods can be broadly categorized into two families: gradient-based methods (Yu et al., 2020; Sener & Koltun, 2018; Navon et al., 2022) and loss-based methods (Liu et al., 2019; Guo et al., 2018). Gradient-based methods utilize aggregation heuristics to combine task gradients into a suitable joint direction. However, these methods often lack stochasticity (Baijiong et al., 2021; Chen et al., 2020; Liu & Vicente, 2024; Xin et al., 2022) and possess a larger convergence set (Kurin et al., 2022) compared to conventional optimization, leading to an increased risk of getting stuck in poor local minima. Conversely, loss-based methods apply direct transformations to the task losses (Baijiong et al., 2021; Kendall et al., 2018) or exploit loss-level information—such as convergence rates (Liu et al., 2019) or task difficulty (Guo et al., 2018)—to guide optimization. While these approaches are often more intuitive, they generally underperform compared to gradient-based methods, as they do not directly address gradient conflicts, which are the primary source of negative transfer.

We propose TaskForce, a novel MTO framework that overcomes the limitations of existing approaches by leveraging cooperative Multi-Agent Reinforcement Learning (MARL) (Lowe et al., 2017). TaskForce frames the MTO problem as a cooperative Markov game (Littman, 1994), where each task-specific agent learns to select aggregation weights for task gradients to minimize the overall loss most effectively. Our approach learns appropriate policies by effectively combining gradient-based and loss-based methods to adapt to the current optimization state.

To address the excessive computational cost of feeding high-dimensional task gradients $\mathbf{g} \in \mathbb{R}^{T \times |\theta|}$ directly into agents, as in standard gradient-based methods, TaskForce instead represents them using the Gram matrix of task gradients, $\mathbf{g}\mathbf{g}^\top \in \mathbb{R}^{T \times T}$. Since the number of tasks $T \ll |\theta|$ is far smaller than the dimensionality of the task gradient, this representation makes the training of multi-agent reinforcement learning computationally feasible. Moreover, it preserves essential optimization signals: the diagonal entries capture the magnitude of each task's gradient, while the off-diagonal entries encode pairwise alignment between tasks. By leveraging this compact yet informative structure, TaskForce enables scalable agent training (see Section E in the appendix) while maintaining the crucial information required to resolve gradient conflicts.

To effectively guide our agents toward learning desirable policies within this novel MARL framework, we carefully construct a reward function that strategically integrates the strengths of established gradient- and loss-based methodologies. Specifically, it combines well-established convex minimization objectives commonly used in provably convergent gradient-based methods (Désidéri, 2012; Sener & Koltun, 2018) with loss convergence rates drawn from loss-based methods (Liu et al., 2019; Guo et al., 2018). By maximizing these rewards, each agent learns to resolve the gradient conflict and scale dominance problem and cooperatively determine update directions, while effectively minimizing the losses across all tasks. As a result, TaskForce bridges the gap between MARL and existing MTO schemes, enabling more effective and robust multi-task learning. We summarize our main contributions as follows:

- We propose TaskForce, a novel multi-task optimization framework that adaptively combines task gradients by using cooperative MARL policies.

- We design a compact yet expressive agent observation based on the Gram matrix of task gradients, capturing both magnitude and pairwise alignment with minimal overhead.

- We introduce a hybrid reward function to leverage both gradient-based and loss-based multi-task optimization for effective update strategies.

- Our method outperforms strong baselines across indoor, outdoor, and molecular benchmarks, demonstrating robust generalization across varied loss types, gradient scales, and task interactions.

## 2 RELATED WORK

### 2.1 MULTI-TASK OPTIMIZATION

General multi-task optimization methods (Sener & Koltun, 2018; Yu et al., 2020; Navon et al., 2022; Senushkin et al., 2023) formulate the MTL training process as a parameterized multi-objective optimization (MOO) problems and aim to directly address the gradient conflict and scale dominance problem that arise during joint training.

**Gradient-based methods:** A prominent category within this paradigm is gradient-based methods (Sener & Koltun, 2018; Chen et al., 2020; Yu et al., 2020; Liu et al., 2021a; Navon et al., 2022), which aggregate task-specific gradients into a unified update direction. These methods attempt to mitigate gradient conflicts by projecting gradients into conflict-free subspaces (Yu et al., 2020; Liu et al., 2021a), reweighting gradients to balance task influence (Navon et al., 2022), or seeking Pareto-stationary solutions in the gradient space (Sener & Koltun, 2018). Most of these approaches are provably convergent and effective in many MTL scenarios, but due to their limited capacity for exploration and reliance on heuristic aggregation rules, they can still converge to suboptimal solutions (Kurin et al., 2022; Xin et al., 2022), particularly under high-conflict conditions.

**Loss-based methods:** Loss-based methods (Baijiong et al., 2021; Liu et al., 2019; Kendall et al., 2018; Guo et al., 2018) take a different approach by modifying the loss functions themselves. This includes reweighting task losses (Baijiong et al., 2021; Kendall et al., 2018) or leveraging additional loss-level signals such as convergence rates (Liu et al., 2019) and task difficulty (Guo et al., 2018). However, these methods are myopic because they do not leverage the gradient-level information, leading to suboptimal results compared to the gradient-based methods.

**Hybrid methods:** More recently, hybrid methods (Liu et al., 2021b; Senushkin et al., 2023; Lin et al., 2023) have emerged, combining both loss-level and gradient-level signals to guide multi-task optimization more holistically. These approaches demonstrate that incorporating both levels of information can effectively reduce scale dominance and mitigate gradient conflict, improving overall optimization performance. Similar to gradient-based methods, these existing techniques also face the problem of potentially becoming stuck in local minima due to their reliance on deterministic heuristic weighting policies. Our methodology can reduce this risk by leveraging the stochasticity inherent in the exploration processes of MARL to enhance the chances of escaping local minima.

## 2.2 REINFORCEMENT LEARNING

Reinforcement learning (RL) (Sutton et al., 1998) has shown significant success in sequential decision-making problems by learning policies that maximize long-term rewards through trial and error. As many real-world applications involve multiple agents, multi-agent reinforcement learning (MARL) (Lowe et al., 2017; Foerster et al., 2018; Gupta et al., 2017) has emerged as a prominent area of research. A key challenge in MARL is the non-stationarity (Lowe et al., 2017) introduced by simultaneously learning agents, which breaks the Markov assumption and hinders convergence. To mitigate this problem, multi-agent deep deterministic policy gradient (MADDPG) (Lowe et al., 2017) extends DDPG (Lillicrap et al., 2015) to the multi-agent setting. This method leverages the centralized training with decentralized execution (CTDE) by equipping each agent with a centralized critic that has access to the observations and actions of all agents. This setup improves training stability and enables agents to learn cooperative policies in both cooperative and mixed settings.

On the other hand, among the multi-task optimization literature, IGBv2 (Dai et al., 2023) is the first to attempt to use single-agent RL to balance the loss weights. However, this method still operates solely at the loss level and, like other loss-based approaches (Baijiong et al., 2021; Liu et al., 2019; Guo et al., 2018), fails to explicitly account for gradient-level conflicts and dominance, limiting the overall performance. In contrast, our approach introduces a multi-task optimization framework leveraging cooperative MARL that directly considers both gradient and loss signals when determining gradient aggregation strategies.

## 3 PRELIMINARIES

### 3.1 GENERAL MULTI-TASK OPTIMIZATION FOR MTL

Given $N$ data points $\{\mathbf{x}^i, \mathbf{y}_1^i, \cdots, \mathbf{y}_T^i\}_{1 \leq i \leq N}$, where $\mathbf{x}^i \in \mathbf{X}$ and $\mathbf{y}_t^i \in \mathbf{Y}_t$ are input data and label collection of $T$ tasks, respectively, the goal of general multi-task optimization is to find the optimal parameters $\theta^*$ of network $\mathcal{F}(\cdot; \theta)$ that minimizes empirical losses. Suppose that there are $t$-th task loss function $\bar{\mathcal{L}}_t(\cdot, \cdot) : \mathbf{Y}_t \times \mathbf{Y}_t \to \mathbb{R}^+$, we can define the $t$-th empirical loss $\mathcal{L}_t(\theta)$ as follows:

$$\mathcal{L}_t(\theta) := \frac{1}{N} \sum_{i=1}^{N} \bar{\mathcal{L}}_t(\mathcal{F}(\mathbf{x}^i; \theta), \mathbf{y}_t^i). \tag{1}$$

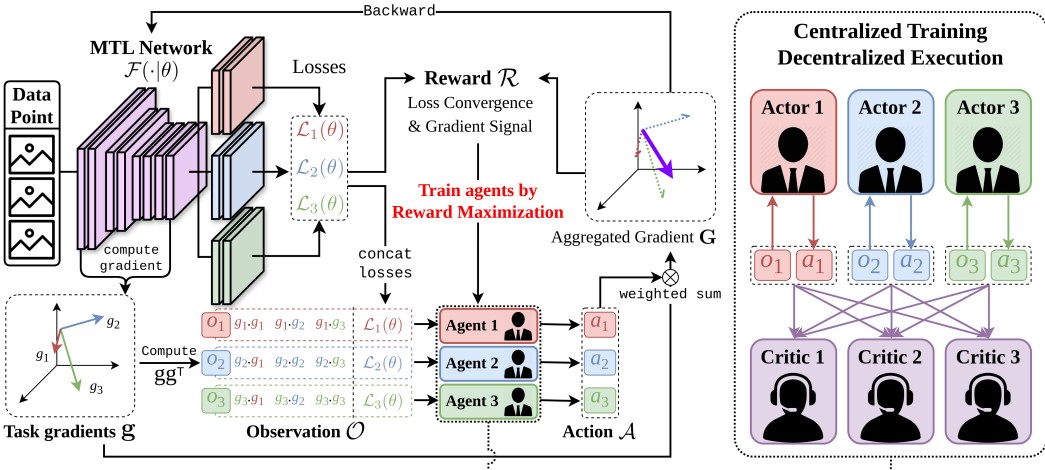

Figure 1: Overall pipeline of TaskForce. Each agent observes task-specific loss and a compact gradient summary via the Gram matrix, predicts a balancing weight for its task gradient, and is guided by a hybrid reward signal that reflects both gradient alignment and loss reduction. Centralized training, decentralized execution allows to learn coordinated policies while reducing computational efficiency by combining global training signals with local, task-specific decision-making.

Then, we can form the given empirical risk minimization into the following multi-objective optimization:

$$\min_{\theta} (\mathcal{L}_1(\theta), \cdots, \mathcal{L}_T(\theta))^{\top}. \tag{2}$$

**Gradient-based Methods** Gradient-based multi-task optimization schemes typically begin by computing the task gradient set $\mathbf{g} = \{g_1, \cdots, g_T\}$ from the empirical loss set $\mathcal{L}(\theta) = \{\mathcal{L}_1(\theta), \cdots, \mathcal{L}_T(\theta)\}$, where $g_t = \nabla_{\theta} \mathcal{L}_t(\theta)$ for each task $t \in \{1, \cdots, T\}$. Each gradient-based method employs its own gradient aggregation algorithm $\Gamma : \mathbb{R}^{|\theta| \times T} \to \mathbb{R}^{|\theta|}$ to compute an aggregated gradient $\mathbf{G}$. Consequently, the model parameters are then updated as follows:

$$\theta \leftarrow \theta - \eta \mathbf{G}, \quad \text{where} \quad \mathbf{G} = \Gamma(g_1, g_2, \cdots, g_T). \tag{3}$$

**Loss-based Methods** In contrast, loss-based methods do not explicitly compute task-wise gradients. Instead, they rely on a loss aggregation algorithm $\Lambda : \mathbb{R}^T \to \mathbb{R}$ that combines the empirical task losses into a single weighted loss $\mathbf{L}(\theta)$. The parameters are then updated by taking the gradient of this aggregated loss as follows:

$$\theta \leftarrow \theta - \eta \nabla_{\theta} \mathbf{L}(\theta), \quad \text{where} \quad \mathbf{L}(\theta) = \Lambda(\mathcal{L}_1(\theta), \mathcal{L}_2(\theta), \cdots, \mathcal{L}_T(\theta)). \tag{4}$$

## 4 METHOD

Similar to other optimization methods, the proposed TaskForce aims to find the optimal parameter $\theta^*$ of the MTL model $\mathcal{F}(\cdot; \theta)$ that minimizes the empirical risk as defined in Equation 2, by leveraging a MARL framework. To this end, TaskForce learns a policy that performs effective gradient aggregation—similar to conventional multi-task optimization methods—based on the empirical loss set $\mathcal{L}$ and task gradients $\mathbf{g}$ computed from each mini-batch.

Realizing this goal requires a precise problem definition and implementation. We begin by casting the multi-task optimization problem as a cooperative Markov game, wherein the game's core components are specifically adapted for this new context (Section 4.1). Subsequently, we leverage this environment to train task-wise agents that learn a cooperative policy to steer the main network's optimization process (Section 4.2). The overall pipeline of our approach is illustrated in Figure 1.

## 4.1 CORE COMPONENTS DESIGN OF MARL

To tightly couple MARL with multi-task optimization, we formulate a cooperative Markov game (Littman, 1994) in which the MTL model itself serves as the interactive, evolving environment, and we assign an individual agent to each task to collaboratively optimize multiple tasks. In the following, we elaborate on how we design the three essential components—(1) observations $\mathcal{O}$, (2) actions $\mathcal{A}$, and (3) rewards $\mathcal{R}$—to formulate a Markov game within the MARL framework.

**Observation** $\mathcal{O}$: Following typical multi-task optimization settings, we construct the observation using the **empirical loss set** $\mathcal{L}(\theta) \in \mathbb{R}^{T \times 1}$ and the corresponding **task gradient set** $\mathbf{g} \in \mathbb{R}^{T \times |\theta|}$, both derived from the $i$-th data point $\{\mathbf{x}^i, \mathbf{y}_1^i, \cdots, \mathbf{y}_T^i\}$ using Equation 1. However, since the gradient dimension $|\theta|$ directly scales with the MTL model's parameters, passing the raw gradients to the reinforcement learning agents would incur significant computational cost (see Table 3). To address this, we leverage the Gram matrix of the task gradient set $\mathbf{g}\mathbf{g}^\top$ and define the observation as:

$$\mathcal{O} = \left\{ \begin{array}{c} o_1 \\ \vdots \\ o_T \end{array} \right\} = \{\mathbf{g}\mathbf{g}^\top | \mathcal{L}(\theta)\} = \left\{ \begin{array}{ccc|c} g_1 \cdot g_1 & \cdots & g_1 \cdot g_T & \mathcal{L}_1(\theta) \\ \vdots & \ddots & \vdots & \vdots \\ g_T \cdot g_1 & \cdots & g_T \cdot g_T & \mathcal{L}_T(\theta) \end{array} \right\}. \tag{5}$$

Our novel agent's observation $o_t \in \mathbb{R}^{T+1}$, constructed from the Gram matrix $\mathbf{g}\mathbf{g}^\top$ and loss $\mathcal{L}$, offers a significantly more compact representation than the complete set of empirical losses and task gradients $\{\mathbf{g} | \mathcal{L}\} \in \mathbb{R}^{T \times (\theta+1)}$, since $T \ll |\theta|$. This observation exclusively encapsulates the agent's local gradient magnitude and its alignment with other task gradients, thereby facilitating efficient, localized decision-making.

**Action** $\mathcal{A}$: The primary objective of each task-specific agent is to infer a balancing parameter that determines its contribution to the final gradient update. We follow the convex combination scheme (Boyd & Vandenberghe, 2004) commonly used in existing gradient-based methods, and obtain the aggregated gradient $\mathbf{G}$ as follows:

$$\mathbf{G} = \sum_{t=1}^{T} w_t g_t, \quad w_t = \frac{\exp(a_t)}{\sum_{k=1}^{T} \exp(a_k)}, \quad a_t = \mu_t(o_t; \phi_t) \in \mathcal{A}, \tag{6}$$

where $\mathcal{A} = \{a_1, \cdots, a_T\}$ denotes the continuous action set from each agent's policy network $\mu_t(\cdot; \phi_t)$, and $w_t \in \mathbb{R}^+$ is the normalized weight obtained via the softmax function to ensure the convexity constraint. Similar to other weight balancing methods (Sener & Koltun, 2018; Liu et al., 2021b), this formulation allows the final update direction to lie within the convex hull of the task gradients, enabling flexible gradient mixing while requiring each agent to output only a constant.

**Next Observation** $\mathcal{O}'$: To reflect the effect of the action $\mathcal{A}$ on the MTL model $\mathcal{F}(\cdot|\theta)$, we first perform a single gradient descent step to update the network parameters as $\theta' \leftarrow \theta - \eta\mathbf{G}$. After that, two main strategies exist to define the next observation. The first strategy uses the same $i$-th data point $\{\mathbf{x}^i, \mathbf{y}_1^i, \cdots, \mathbf{y}_T^i\}$ to compute a new empirical loss set $\mathcal{L}(\theta')$ and the corresponding task gradient set $\mathbf{g}'$. These are then used to generate the next observation via equation 5. However, this approach incurs significant computational overhead, requiring two forward and backward passes for every update step. The second strategy utilizes the subsequent $(i + 1)$-th data point $\{\mathbf{x}^{i+1}, \mathbf{y}_1^{i+1}, \cdots, \mathbf{y}_T^{i+1}\}$ to generate the next observation $\mathcal{O}'$. This method obviates the need for additional forward-backward steps, which are inherent to the first strategy. Consequently, it significantly reduces training time compared to the first strategy, and thus, we adopt this latter approach.

**Reward** $\mathcal{R}$: One key strength of reinforcement learning is its flexibility in handling non-differentiable and highly controllable reward functions. We leverage this to design a reward that combines both loss-based feedback for immediate progress and gradient-based signals for long-term task balancing. In our MTL setting, RL agents are trained alongside the shared network $\mathcal{F}$, and must quickly adapt at each training step. To provide immediate feedback, we define a simple loss-based reward that measures the relative improvement in log-transformed task losses:

$$r_{\mathcal{L}} = \sum_{t=1}^{T} \log(1 + \mathcal{L}_t(\theta)) - \sum_{t=1}^{T} \log(1 + \mathcal{L}_t(\theta')), \tag{7}$$

where $\mathcal{L}_t(\theta)$ and $\mathcal{L}_t(\theta')$ represent the $t$-th empirical loss before and after the update, respectively. Note that the logarithmic transform provides scale-invariant measurement of loss improvement (Navon et al., 2022; Lin et al., 2023), making the reward more robust across different scales of task losses.

While $r_{\mathcal{L}}$ captures per-iteration loss convergence progress, it does not consider interactions between gradients, which are known to affect general MTL performance significantly. To address this, we design a gradient-based reward term $r_{\mathcal{G}}$ that evaluates the value of the aggregated gradient $\mathbf{G}$. Specifically, $r_{\mathcal{G}}$ leverages a convex minimization problem (Désidéri, 2012; Sener & Koltun, 2018), widely used in multi-objective optimization to find a common descent direction that simultaneously minimizes all objective functions and converges to a Pareto optimal point.

$$\underset{w_1,\cdots,w_T}{\text{minimize}} \ \| \sum_{t=1}^{T} w_t g_t \|_2^2, \quad \text{subject to} \ \sum_{t=1}^{T} w_t = 1, \ w_t \geq 0. \tag{8}$$

Our gradient-based reward $r_{\mathcal{G}}$ reformulates the convex minimization problem into a reward-level maximization problem suitable for a Markov game as follows:

$$r_{\mathcal{G}} = -\| \sum_{t=1}^{T} w_t g_t \|_2^2 = -\| \sum_{t=1}^{T} \mathbf{G} \|_2^2. \tag{9}$$

This allows the agent to learn and select a policy that aligns with a provably convergent direction, thereby improving the overall stability and performance of the multi-objective learning process.

The final reward used for policy learning is a weighted sum of these components:

$$\mathcal{R} = \lambda_{\mathcal{L}} r_{\mathcal{L}} + \lambda_{\mathcal{G}} r_{\mathcal{G}}, \tag{10}$$

where $\lambda_{\mathcal{L}}$ and $\lambda_{\mathcal{G}}$ are hyperparameters that control the trade-off between per-iteration loss improvement $r_{\mathcal{L}}$ and desirable gradient property $r_{\mathcal{G}}$ which is related to Pareto convergence. Note that the reward $\mathcal{R}$ is shared by all agents due to the fully cooperative scenario.

## 4.2 TRAINING OF TASKFORCE

In this section, we introduce the training procedure of the proposed TaskForce framework. For the multi-agent reinforcement learning algorithm, we adopt Lowe et al. (2017). We detail the systematic process of this framework in the subsequent discussion. Consider a Markov game in a multi-agent setting, where $T$ task-wise agents interact with a shared environment. The $t$-th agent receives a local observation $o_t$ and selects an action $a_t$ according to its policy $\mu_t(o_t; \phi_t)$.

Following the centralized training and decentralized execution paradigm, each agent is equipped with a decentralized policy $\mu_t(\cdot; \phi_t)$ and a centralized critic $Q_t^{\boldsymbol{\mu}}(\cdot, \cdot; \psi_t)$. In the off-policy training, agents are trained using a replay buffer $\mathbf{D}$ containing transitions $(\mathcal{O}, \mathcal{A}, \mathcal{R}, \mathcal{O}')$, which are collected in advance by executing the current policies $\boldsymbol{\mu}$ jointly with the multi-task model in the environment.

First, the critic is trained to minimize the temporal difference (TD) loss as follows:

$$\mathcal{L}(\psi_t) = \mathbb{E}_{(\mathcal{O},\mathcal{A},\mathcal{R},\mathcal{O}')\sim\mathbf{D}} \left[ \left( Q_t^{\boldsymbol{\mu}}(\mathcal{O},\mathcal{A}; \psi_t) - (\mathcal{R} + \gamma Q_t^{\boldsymbol{\mu}'}(\mathcal{O}',\mathcal{A}'; \psi_t')) \right)^2 \right],$$
$$a_t' = \mu_t(o_t'; \phi_t') \in \mathcal{A}' \ \ \forall 1 \leq t \leq T, \tag{11}$$

where the target value is computed using the set of target policies $\boldsymbol{\mu}' = \{\mu_1(\cdot; \phi_1'), \cdots, \mu_T(\cdot; \phi_T')\}$ and target critic $Q_t^{\boldsymbol{\mu}'}$ with delayed parameters $(\phi_t', \psi_t')$, and $\gamma$ is the discount factor, respectively. The actor is then updated via the deterministic policy gradient as follows:

$$\nabla_{\phi_t} J(\phi_t) = \mathbb{E}_{(\mathcal{O},\mathcal{A})\sim\mathbf{D}} \left[ \nabla_{\phi_t}\mu_t(o_t; \phi_t) \nabla_{a_t} Q_t^{\boldsymbol{\mu}}(\mathcal{O},\mathcal{A}; \psi_t)\big|_{a_t=\mu_t(o_t;\phi_t)} \right]. \tag{12}$$

Lastly, target networks are updated using soft updates with an exponential moving average coefficient $0 < \tau \ll 1$ as follows:

$$\phi_t' \leftarrow \tau\phi_t + (1-\tau)\phi_t', \quad \psi_t' \leftarrow \tau\psi_t + (1-\tau)\psi_t'. \tag{13}$$

---

**Algorithm 1** Training Process of TaskForce.

---

**Input:** data point number $N$, task number $T$, data points $\mathbf{X}, \{\mathbf{Y}_t\}_{1 \leq t \leq T}$, MTL model $\mathcal{F}(\cdot; \theta)$, agents $\{\mu_t(\cdot; \phi_t), Q_t^{\boldsymbol{\mu}}(\cdot, \cdot; \psi_t)\}_{1 \leq t \leq T}$, replay buffer $\mathbf{D}$, batch size of agents $b_{\text{agent}}$.
**Output:** trained MTL model $\mathcal{F}(\cdot; \theta^*)$, trained agents $\{\mu_t(\cdot; \phi_t^*), Q_t^{\boldsymbol{\mu}}(\cdot, \cdot; \psi_t^*)\}_{1 \leq t \leq T}$.

1: initialize $\mathcal{O}_{\text{prev}}, \mathcal{A}_{\text{prev}}, \mathcal{R}_{\text{prev}}$ as null matrix or 0.
2: **for** $i = 1$ **to** $N$ **do**
3:     sample data point $\{\mathbf{x}^i, \mathbf{y}_1^i, \cdots, \mathbf{y}_T^i\}$.
4:     compute empirical loss set $\boldsymbol{\mathcal{L}}(\theta)$ and task gradient set $\mathbf{g}$ from data point.
5:     generate observation $\mathcal{O}$ from $\boldsymbol{\mathcal{L}}(\theta), \mathbf{g}\mathbf{g}^\top$ by Equation 5.
6:     compute action $\mathcal{A} = \{\mu_1(o_1; \phi_1), \cdots, \mu_T(o_T; \phi_T)\}$ from $\mathcal{O}$.
7:     compute aggregated gradient $\mathbf{G}$ from $\mathbf{g}$ and $\mathcal{A}$ by Equation 6.
8:     **if** $i \neq 1$ **then**
9:         compute reward $\mathcal{R}$ from $\boldsymbol{\mathcal{L}}(\theta), \boldsymbol{\mathcal{L}}_{\text{prev}}$ by Equation 7-10.
10:        push transition $(\mathcal{O}_{\text{prev}}, \mathcal{A}_{\text{prev}}, \mathcal{R}_{\text{prev}}, \mathcal{O})$ to replay buffer $\mathbf{D}$.
11:    **end if**
12:    $\mathcal{O}_{\text{prev}}, \mathcal{A}_{\text{prev}}, \mathcal{R}_{\text{prev}}, \boldsymbol{\mathcal{L}}_{\text{prev}} \leftarrow \mathcal{O}, \mathcal{A}, \mathcal{R}, \boldsymbol{\mathcal{L}}(\theta)$.
13:    update MTL model $\mathcal{F}(\cdot | \theta)$ with $\mathbf{G}$ by $\theta \leftarrow \theta - \eta \mathbf{G}$.
14:    **if** $i > b_{\text{agent}}$ **then**
15:        sample $b_{\text{agent}}$ transitions $\mathcal{T}$ from replay buffer $\mathbf{D}$.
16:        update each actor $\mu_t(\cdot; \phi_t)$ and critic $Q_t^{\boldsymbol{\mu}}(\cdot, \cdot; \psi_t)$ with transitions $\mathcal{T}$ by Equation 11-13.
17:    **end if**
18: **end for**

---

Algorithm 1 summarizes the training process of TaskForce. First, to enable off-policy training with the replay buffer $\mathbf{D}$, we compute empirical losses and their gradients from a data point and store them as transition tuples $(\mathcal{O}, \mathcal{A}, \mathcal{R}, \mathcal{O}')$. Second, the MTL model parameters are updated using the aggregated gradient $\mathbf{G}$, derived from the agents' actions $\mathcal{A}$. Finally, the agent parameters are updated by sampling transitions from the replay buffer, completing one MTL training iteration. For clarity, we omit implementation details such as exploration noise scaling and reward normalization.

## 5 EXPERIMENTS

**Baselines** Similar to previous works (Navon et al., 2022; Senushkin et al., 2023), we compare our TaskForce with the well-known multi-task optimization approaches: (1) Linear Scalarization (LS) which minimizes $\sum_{t=1}^{T} \mathcal{L}_t(\theta)$; (2) Random Loss Weighting (RLW) (Baijiong et al., 2021); (3) Dynamic Weight Average (DWA) (Liu et al., 2019); (4) Uncertainty Weighting (UW) (Kendall et al., 2018); (5) Multiple Gradient Descent Algorithm (MGDA) (Sener & Koltun, 2018); (6) GradDrop (Chen et al., 2020); (7) PCGrad (Yu et al., 2020); (8) CAGrad (Liu et al., 2021a); (9) Improvable Gap Balancing (IGBv2) (Dai et al., 2023); (10) IMTL (Liu et al., 2021b); (11) NashMTL (Navon et al., 2022); (12) Aligned-MTL (Senushkin et al., 2023).

**Datasets & Model Architecture** *NYU-v2* (Silberman et al., 2012) is an indoor scene understanding benchmark with 795 training and 654 testing samples, annotated for three tasks: 13-class semantic segmentation, depth estimation, and surface normal estimation. We use the MTAN (Liu et al., 2019) architecture for evaluation. *Cityscapes* (Cordts et al., 2016) focuses on urban scene understanding and provides 2,975 training and 500 validation images from 50 cities. It supports three tasks: 7-class semantic segmentation, instance segmentation, and depth estimation. We adopt PSPNet (Zhao et al., 2017) for evaluation. *QM9* (Ramakrishnan et al., 2014) is a molecular property prediction dataset with 110K training, 10K validation, and 10K test molecules. It covers 11 regression tasks, each predicting a quantum chemical property. We use the MPNN (Gilmer et al., 2017) architecture.

**Metrics & Experimental Setup** We follow the task-specific evaluation metrics used in (Navon et al., 2022) for the NYU-v2 and QM9 datasets, and those in (Senushkin et al., 2023) for the Cityscapes dataset. To assess the overall performance across different metrics and tasks, we adopt the relative performance decrement measures $\Delta \mathbf{m}$ and $\Delta \mathbf{t}$—Formally, $\Delta \mathbf{m}$ is defined as $\Delta \mathbf{m} = 1/K \sum_{k=1}^{K} (-1)^{\delta_k} (M_{\text{MTL},k} - M_{\text{STL},k}) / M_{\text{STL},k}$, where $K$ is the number of metrics, $M_{\text{STL},k}$ and $M_{\text{MTL},k}$ represent the $k$-th metric for the STL and MTL models, respectively. The indicator $\delta_k$ equals 1 if a higher value is better for the $k$-th metric and 0 otherwise. The overall task-level perfor-

Table 1: Evaluation results of NYU-v2 3-tasks setup. We report MTAN Liu et al. (2019) model performance averaged over 3 random seeds.

| Method | Semseg. | | Depth | | Normal | | | | | $\Delta$m $\downarrow$ | $\Delta$t $\downarrow$ |
|---|---|---|---|---|---|---|---|---|---|---|---|
| | mIoU | PAcc. | Abs. | Rel. | Mean | Median | 11.25° | 22.5° | 30° | | |
| STL | 38.30 | 63.76 | 0.68 | 0.28 | 25.01 | 19.21 | **30.14** | **57.20** | 69.15 | 0.00% | 0.00% |
| LS | 39.29 | 65.33 | 0.55 | 0.23 | 28.15 | 23.96 | 22.09 | 47.50 | 61.08 | +5.46% | −1.07% |
| RLW | 37.17 | 63.77 | 0.58 | 0.24 | 28.27 | 24.18 | 22.26 | 47.05 | 60.62 | +7.67% | +2.00% |
| DWA | 39.11 | 65.31 | 0.55 | 0.23 | 27.61 | 23.18 | 24.17 | 50.18 | 62.39 | +3.49% | −2.06% |
| UW | 36.87 | 63.17 | 0.54 | 0.23 | 27.04 | 22.61 | 23.54 | 49.05 | 63.65 | +4.01% | −0.97% |
| MGDA | 30.47 | 59.90 | 0.61 | 0.26 | 24.88 | 19.45 | 29.18 | 56.88 | **69.36** | +1.47% | +1.79% |
| GradDrop | 39.39 | 65.12 | 0.55 | 0.23 | 27.48 | 22.96 | 23.38 | 49.44 | 62.87 | +3.61% | −2.03% |
| PCGrad | 38.06 | 64.64 | 0.56 | 0.23 | 27.41 | 22.80 | 23.86 | 49.83 | 63.14 | +3.83% | −1.33% |
| CAGrad | 39.79 | 65.49 | 0.55 | 0.23 | 26.31 | 21.58 | 25.61 | 52.36 | 65.58 | +0.29% | −4.18% |
| IGBv2 | 38.53 | 64.81 | 0.55 | 0.23 | 26.54 | 22.11 | 24.90 | 52.21 | 66.09 | +1.71% | −2.61% |
| IMTL | 39.35 | 65.60 | 0.54 | 0.23 | 26.02 | 21.19 | 26.20 | 53.13 | 66.24 | −0.59% | −4.76% |
| NashMTL | 40.13 | 65.93 | 0.53 | **0.22** | 25.26 | 20.08 | 28.40 | 55.47 | 68.15 | −4.04% | −7.56% |
| Aligned-MTL | 40.82 | 66.33 | 0.53 | **0.22** | 25.19 | 19.71 | 28.88 | 56.23 | 68.54 | −4.93% | −8.40% |
| TaskForce (Ours) | **41.77** | **66.73** | **0.51** | **0.22** | **24.83** | **19.19** | 29.27 | 56.85 | 69.29 | **−6.47%** | **−9.96%** |

Table 2: Evaluation results on Cityscapes (3-tasks) and QM9 (11-tasks) setups. We report model performance averaged over 3 random seeds for PSPNet (Cityscapes) and MPNN (QM9).

| Method | **Cityscapes** | | | | **QM9** |
|---|---|---|---|---|---|
| | Semseg. mIoU (%) $\uparrow$ | Instseg. L1 (px.) $\downarrow$ | Disparity MSE $\downarrow$ | $\Delta$m $\downarrow$ | $\Delta$m $\downarrow$ |
| STL | 66.73 | 10.55 | 0.33 | 0.00% | 0.00% |
| LS | 52.98 | 10.89 | 0.39 | +14.30% | +177.6% |
| RLW | 51.26 | 10.25 | 0.41 | +15.58% | +203.8% |
| DWA | 53.15 | 10.22 | 0.40 | +13.20% | +175.3% |
| UW | 60.12 | **9.87** | 0.33 | +1.53% | +108.0% |
| MGDA | 66.72 | 17.02 | 0.33 | +20.62% | +120.5% |
| GradDrop | 52.98 | 10.09 | 0.40 | +12.50% | +198.7% |
| PCGrad | 54.06 | 9.91 | 0.38 | +10.00% | +125.7% |
| CAGrad | 64.33 | 10.15 | 0.34 | +1.46% | +112.8% |
| IGBv2 | 61.14 | 10.53 | 0.33 | +2.73% | +67.7% |
| IMTL | 65.13 | 11.58 | **0.32** | +3.10% | +77.2% |
| NashMTL | 64.84 | 11.90 | 0.37 | +9.38% | +62.0% |
| Aligned-MTL | **67.06** | 10.63 | 0.33 | −0.02% | +81.9% |
| TaskForce (Ours) | 66.63 | 10.55 | **0.32** | **−0.65%** | **+59.0%** |

mance decrement, $\Delta$t, is computed as the average of $\Delta$m across all tasks. We follow the training protocol of (Senushkin et al., 2023) for NYU-v2 and Cityscapes, and (Navon et al., 2022) for QM9, respectively. Across all experiments, we set $\lambda_{\mathcal{L}} = 1.0$ for the loss-based reward, $\lambda_{\mathcal{G}} = 1 \times 10^{-3}$ for the gradient-based reward. We adopt a standard MADDPG (Lowe et al., 2017) for agents. Due to the space constraints, we describe the implementation details of the RL agents in Appendix Section B.

**Evaluation Results on NYU-v2 & Cityscapes** Table 1 and Table 2 summarize the performance of TaskForce against baseline methods on the NYU-v2 and Cityscapes datasets, respectively. Across both benchmarks, TaskForce consistently outperforms all baselines according to the relative performance measures $\Delta$m and $\Delta$t. Notably, our method shows consistent improvement over the strong competitor, Aligned-MTL, across nearly all reported metrics, with the sole exception being the segmentation on the Cityscapes dataset. This robust performance demonstrates that our method successfully navigates the complex multi-task landscape, empowered by its cooperative MARL setup.

**Evaluation Results on QM9** QM9 is one of the most challenging datasets in multi-task learning due to its complex and diverse molecular properties across tasks, and the significant scale difference between losses. As shown in Table 2, TaskForce significantly outperforms all competing baseline methods. We observe that strong competitors like Aligned-MTL, while performing well on simpler scene understanding datasets, struggle considerably under the high task complexity of QM9. This outcome demonstrates the resilience of the proposed method, which performs robustly even when faced with an increased number of tasks and more complex gradient interactions.

Table 3: Ablation studies on Cooperative MARL components on the NYU-v2 3-tasks setup. We report the training cost relative to the final configuration of each method on the MTAN Liu et al. (2019) network architecture. We set $\mathcal{R} = r_{\mathcal{L}}$ of all configuration except $r_{\mathcal{G}}$ ablation. (MA: multi-agents, CT: centralized training, DE: decentralized execution, *: rough calculation.)

| $\mathbf{gg}^\top$ | MA | CT | DE | $r_{\mathcal{G}}$ | training cost | $\Delta\mathbf{m}\downarrow$ | $\Delta\mathbf{t}\downarrow$ |
|---|---|---|---|---|---|---|---|
| | | | | | $\times 2.59\text{M}^*$ | - | - |
| ✓ | | | | | $\times 0.95$ | $-2.89\%$ | $-4.05\%$ |
| ✓ | ✓ | | | | $\times 1.16$ | $-4.26\%$ | $-7.19\%$ |
| ✓ | ✓ | ✓ | | | $\times 3.21$ | $-5.23\%$ | $-8.31\%$ |
| ✓ | ✓ | ✓ | ✓ | | $\times 1.00$ | $-5.18\%$ | $-8.26\%$ |
| ✓ | ✓ | ✓ | ✓ | ✓ | $\times 1.00$ | $\mathbf{-6.47\%}$ | $\mathbf{-9.96\%}$ |

## 5.1 Ablation Studies on Cooperative MARL Components

To evaluate the contribution of each component of our cooperative multi-agent reinforcement learning framework within TaskForce, we conduct ablation studies by incrementally introducing five key elements: (1) gram-matrix observation ($\mathbf{gg}^\top$); (2) task-specific multiple agents (MA); (3) a centralized critic training (CT) that processes joint observations and actions; (4) decentralized execution (DE); (5) shared $\mathcal{R}$ based on the loss reduction rate across all tasks, instead of each loss reduction rewards (We detail the experimental setup of this ablation in Appendix section A).

As shown in Table 3, we observe consistent performance gains as each key component of cooperative MARL is added. Due to the excessively large number of shared parameter $|\theta| \sim= 44.1M$ of MTAN (Liu et al., 2019), the configuration that does not incorporate the task gradient gram matrix $\mathbf{gg}^\top \in \mathbb{R}^{T \times T}$ is not appropriate for the multi-task optimization framework. The multiple agents allow task-specific specialization, enabling the model to disentangle conflicting optimization signals across tasks. The centralized critic improves credit assignment by leveraging global information, leading to better optimization performance. Meanwhile, decentralized execution, which relies only on task-specific local observations, enhances training efficiency with minimal compromise in overall performance. Lastly, using the gradient-based reward $r_{\mathcal{G}}$ encourages the agents to consistently align with a provably convergent direction, thereby improving performance. These findings highlight that the modular and cooperative structure of MARL, particularly when fully integrated, plays a crucial role in enhancing both convergence stability and overall performance in multi-task learning scenarios. Notably, our TaskForce substantially reduces the computational cost associated with reinforcement learning by leveraging the Gram matrix and decentralized execution (DE), resulting in a training cost that remains comparable to that of conventional methods (See appendix Section E).

## 6 Conclusion

We have introduced TaskForce, a MARL-based framework for multi-task optimization that reformulates MTL as a cooperative Markov game. Unlike prior approaches that aggregate gradients or rely on heuristic balancing, TaskForce models each task as an agent equipped with a compact gradient-based observation and a loss–gradient hybrid reward, enabling cooperative strategies that balance tradeoffs, resolve conflicts, and adaptively guide optimization toward Pareto-efficient solutions. At the core of TaskForce is a lightweight agent observation derived from the Gram matrix of task gradients, capturing both magnitudes and pairwise alignments with minimal computational overhead. Complementing this, we design a principled gradient-based reward grounded in convex multi-objective optimization, which provides theoretical convergence guarantees while promoting cooperative task interactions. Extensive experiments on NYU-v2, Cityscapes, and QM9 demonstrate that TaskForce consistently surpasses strong baselines, yielding more stable convergence, stronger generalization across domains, and improved task-level performance. These results establish TaskForce as an effective bridge between cooperative MARL and gradient-based optimization for multi-task learning. Looking ahead, we envision extending TaskForce to larger and more diverse task sets, incorporating richer reward structures, and applying it to real-world domains in vision, language, and molecular modeling, further expanding the potential of MARL-based optimization in deep learning.

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

# Appendix

## A EXPERIMENTAL SETUP OF ABLATION STUDIES

In the main manuscript, we conducted an ablation study on TaskForce by incrementally integrating five key components: (1) Gram matrix observation ($\mathbf{gg}^\top$), (2) Multi-Agent (MA), (3) Centralized Training (CT), (4) Decentralized Execution (DE), and (5) adding gradient-based reward $r_\mathcal{G}$. This section provides a detailed description of the experimental configurations for each ablation stage, supplementing the explanations in the main manuscript:

1. **Gram matrix ($\mathbf{gg}^\top$)**: If we construct the agent observations $\mathcal{O}$ using the raw task gradient set $\mathbf{g}$ and the empirical loss set $\mathbf{\mathcal{L}}(\theta)$, the observations are as follows:

$$\mathcal{O} = \left\{ \begin{array}{c} o_1 \\ \vdots \\ o_T \end{array} \right\} = \{\mathbf{g}|\mathbf{\mathcal{L}}(\theta)\} = \left\{ \begin{array}{c|c} g_1 & \mathcal{L}_1(\theta) \\ \vdots & \vdots \\ g_T & \mathcal{L}_T(\theta) \end{array} \right\} \in \mathbb{R}^{T \times (|\theta|+1)}, \tag{14}$$

   where $g_t = \nabla_\theta \mathcal{L}_t(\theta)$ for each task $t \in \{1, \cdots, T\}$. Since the number of shared parameters $|\theta|$ is exceedingly large compared to the number of tasks $T$ ($T \ll |\theta|$), this formulation results in an observation space with prohibitive spatial and temporal complexity. Thus, as reported in Table 3, rendering the multi-task optimization process computationally infeasible.

   Therefore, we propose TaskForce that designs the compact observation based on Gram matrix of the task gradient set as follows:

$$\mathcal{O} = \left\{ \begin{array}{c} o_1 \\ \vdots \\ o_T \end{array} \right\} = \{\mathbf{gg}^\top|\mathbf{\mathcal{L}}(\theta)\} = \left\{ \begin{array}{ccc|c} g_1 \cdot g_1 & \cdots & g_1 \cdot g_T & \mathcal{L}_1(\theta) \\ \vdots & \ddots & \vdots & \vdots \\ g_T \cdot g_1 & \cdots & g_T \cdot g_T & \mathcal{L}_T(\theta) \end{array} \right\} \in \mathbb{R}^{T \times (T+1)}. \tag{15}$$

   Note that in setups where Decentralized Execution (DE) is not yet applied, the policy uses the entire observation matrix $\mathcal{O}$ as an input.

2. **Multi-agent training (MA)**: The single-agent baseline utilizes the DDPG (Lillicrap et al., 2015). Its policy network, $\mu(\mathcal{O}; \phi)$, takes the entire observation $\mathcal{O}$ and directly outputs the joint action vector $\mathcal{A}$. Conversely, the multi-agent configuration employs MADDPG (Lowe et al., 2017) as introduced in the main manuscript. We introduce task-specific policies, where each policy $\mu_t(\mathcal{O}; \phi_t)$ takes the shared observation $\mathcal{O}$ but outputs only its corresponding action $a_t$ as follows:

$$\mathcal{A} = [a_1, \cdots, a_T] = [\mu_t(\mathcal{O}; \phi_t)]_{t=1}^T. \tag{16}$$

   In this configuration, each critic $Q_t^\mu$ is still decentralized in its action evaluation; it takes the shared observation $\mathcal{O}$ and only the action $a_t$ from its own agent to estimate the action value $\mathcal{Q}$ as follows:

$$\mathcal{Q} = Q_t^\mu(\mathcal{O}, a_t; \psi_t). \tag{17}$$

3. **Centralized Training (CT)**: A key contribution of MADDPG is its centralized critic, which we introduce in this step. Unlike the previous setup, each critic $Q_t^{\boldsymbol{\mu}}$ now evaluates its action value $\mathcal{Q}$ from the entire action $\mathcal{A}$ of all policies $\boldsymbol{\mu} = \{\mu_1, \cdots, \mu_T\}$, in addition to the shared observation $\mathcal{O}$. This allows for more effective credit assignment during training:

$$\mathcal{Q} = Q_t^\mu(\mathcal{O}, \mathcal{A}; \psi_t). \tag{18}$$

4. **Decentralized Execution (DE)**: To accelerate inference and to improve practicality, many MARL algorithms utilize decentralized execution. In this paradigm, each policy network $\mu_t$ uses only its local, task-specific observation $o_t$ as input, rather than the full observation matrix $\mathcal{O}$. The critic, however, continues centralized training manner during training, utilizing the global observation $\mathcal{O}$ and the joint action $\mathcal{A}$ as follows:

$$\mathcal{A} = [a_1, \cdots, a_T] = [\mu_t(o_t; \phi_t)]_{t=1}^T, \tag{19}$$

$$\mathcal{Q} = Q_t^\mu(\mathcal{O}, \mathcal{A}; \psi_t). \tag{20}$$

5. **Gradient-Based Reward ($r_\mathcal{G}$)**: In the final configuration, we incorporate the proposed gradient-based reward term, $r_\mathcal{G}$. While previous setups relied solely on a loss-based reward, $\mathcal{R} = \lambda_\mathcal{L} r_\mathcal{L}$, the final reward function is a weighted sum of both components:

$$\mathcal{R} = \lambda_\mathcal{L} r_\mathcal{L} + \lambda_\mathcal{G} r_\mathcal{G}. \tag{21}$$

## B  HYPERPARAMETER AND RL AGENT CONFIGURATION OF TASKFORCE

As aforementioned, in our work, we adapt the MADDPG (Lowe et al., 2017) algorithm, where the number of actor and critic networks corresponds to the number of agents (the number of tasks in our method). Each actor and critic network is implemented as a three-layer MLP with a hidden dimension of 64. The agents are trained using the Adam optimizer (Kingma & Ba, 2014) with a learning rate of $5 \times 10^{-4}$ and a transition batch size of 64. The experience replay buffer stores up to 100,000 transitions, allowing the agents to perform off-policy learning effectively. We employ soft target updates (Lillicrap et al., 2015) for stabilizing the critic network, using an exponential moving average coefficient of $\tau = 0.01$. The discount factor for delayed rewards is set to $\gamma = 0.95$. To encourage exploration, we utilize Ornstein-Uhlenbeck (OU) noise (Lillicrap et al., 2015) during training. The noise scale linearly decays from 0.3 to 0.05 over the first 10,000 training iterations. We use the $\log$-transformed empirical loss to ensure scale-invariant conditions across tasks. Also, gradient normalization (Sener & Koltun, 2018) is applied to prevent instability due to the scale dominance problem in the gradient space.

## C  COMPATIBILITY OF GRADIENT-BASED REWARD WITH OTHER CONVEX MINIMIZATION

Beyond the gradient-based reward $r_{\mathcal{G}}$ utilized in this paper, the multi-objective optimization litera-ture presents various approaches for solving convex minimization problems, often grounded in dif-ferent hypotheses (Liu et al., 2021b; Navon et al., 2022). To investigate whether our gradient-based reward framework is compatible with these alternative formulations, we adapt the minimization problem from a strong baseline, IMTL (Liu et al., 2021b).

IMTL encourages a balanced update by equalizing the projection of the aggregated gradient onto each normalized task gradient. This is also framed as a minimization objective, negated into a reward as follows:

$$\underset{w_1, \cdots, w_T}{\text{minimize}} \sum_{t=2}^{T} \|\mathbf{G}(g_1/\|g_1\|_2 - g_t/\|g_t\|_2)^\top\|_2^2, \quad \text{subject to} \sum_{t=1}^{T} w_t = 1, \ w_t \geq 0, \quad (22)$$

$$r_{\mathcal{G}}^{\text{IMTL}} = -\sum_{t=2}^{T} \|\mathbf{G}(g_1/\|g_1\| - g_t/\|g_t\|)^\top\|_2^2. \quad (23)$$

We conduct additional experiments to validate the performance of TaskForce using this IMTL-based reward $r_{\mathcal{G}}^{\text{IMTL}}$, with results presented in Table 7-9. Across all experiments in the appendix, we set $\lambda_{\mathcal{L}} = 1.0$ for the loss-based reward, $\lambda_{\mathcal{G}} = 1 \times 10^{-3}$ for the original gradient-based reward, and $\lambda_{\mathcal{G}}^{\text{IMTL}} = 1.0$ for the IMTL-based reward. As shown in Table 7, when the gradient-based reward is replaced with $r_{\mathcal{G}}^{\text{IMTL}}$, TaskForce maintains its high performance and continues to outperform strong baselines. This result demonstrates the potential extensibility of our methodology, indicating that the gradient-based reward can be effectively constructed from various gradient-level convex minimiza-tion problems proposed in the multi-objective optimization literature.

## D  SENSITIVITY OF REWARD WEIGHT PARAMETERS OF THE TASKFORCE.

Beyond the standard hyperparameters of MADDPG (e.g., replay buffer size, discount factor), our proposed TaskForce introduces two tunable reward weights $\lambda_{\mathcal{L}}$ and $\lambda_{\mathcal{G}}$. Due to the MADDPG utiliz-ing the reward normalization for training stability (Lowe et al., 2017), the relative proportion of $\lambda_{\mathcal{L}}$ and $\lambda_{\mathcal{G}}$ only significantly influences the learning outcome. As specified in the upper implementation details section, we used the same reward weights across all experiments. The relative proportions of these lambda values were determined experimentally to ensure a comparable average scale between the loss reward and gradient reward (i.e., $\lambda_{\mathcal{L}} = 1.0, \lambda_{\mathcal{G}} = 0.001, \lambda_{\mathcal{G}}^{IMTL} = 1.0$). The consistent performance improvements observed across all datasets experimentally demonstrate that our chosen lambda values can generalize well in practice.

Additionally, we conducted a grid search based on the ratio of the reward weight parameters, as shown in Table 4. We incorporate the original gradient-based reward $r_{\mathcal{G}}$ and other gradient-based

reward $r_{\mathcal{G}}^{\text{IMTL}}$ introduced in Section C for this grid search. The results on the Cityscapes dataset show that our method outperforms the current state-of-the-art method, Aligned-MTL (Senushkin et al., 2023) ($\Delta\mathbf{m} = -0.01\%$), in most cases except for experiments where loss-based rewards were not leveraged, or where the gradient reward scale became excessively large (i.e., $\lambda_{\mathcal{L}} : \lambda_{\mathcal{G}} = 1 : 1$).

Table 4: Grid search of reward weight parameters $\lambda_{\mathcal{L}}$ and $\lambda_{\mathcal{G}}$ on the Cityscapes dataset. We report PSPNet Zhao et al. (2017) model performance averaged over 3 random seeds.

| Reward Ratio ($\lambda_{\mathcal{L}} : \lambda_{\mathcal{G}}$) | $\Delta\mathbf{m}$ ($\lambda_{\mathcal{G}}$) | $\Delta\mathbf{m}$ ($\lambda_{\mathcal{G}}^{\text{IMTL}}$) |
|---|---|---|
| 1:0 | $-0.33\,\%$ | $-0.33\,\%$ |
| 1:0.01 | $-0.65\,\%$ | $-0.41\,\%$ |
| 1:1 | $+0.31\,\%$ | $-0.66\,\%$ |
| 0:1 | $+10.42\,\%$ | $+6.10\,\%$ |

# E    COMPUTATIONAL OVERHEAD COMPARISON BETWEEN TASKFORCE AND PRIOR METHODS

To evaluate the computational overhead and training efficiency of our proposed TaskForce—which uses multiple RL agents instead of a traditional numerical solver—we measured the per-epoch wall time for our method and baselines across all datasets. All experimental setups are identical to the implementation details provided in the main manuscript and the Appendix. Note that all computational costs were measured on a single A6000 GPU.

As shown in Table 5, our experimental results indicate that for the 3-task scenario, **there is no significant difference in wall time** compared to existing gradient- and hybrid-based optimization methods. Furthermore, even in the more challenging 11-task quantum chemistry setup, where task complexity causes a scalability issue, our method demonstrates acceptable training time when compared to conventional approaches.

Table 5: Per-epoch training cost (epoch/sec) comparison between the proposed TaskForce and other baselines.

| Method | NYU-v2 3-tasks | Cityscapes 3-tasks | QM9 11-tasks |
|---|---|---|---|
| LS | 85 | 168 | 85 |
| MGDA | 114 | 261 | 332 |
| IMTL | 112 | 258 | 294 |
| NashMTL | 109 | 258 | 286 |
| Aligned-MTL | 111 | 255 | 279 |
| Ours | 111 | 257 | 304 |

To further analyze the computational cost of TaskForce, which requires additional agent learning, we also measured the contribution of each sub-procedure within TaskForce to the overall computational cost. It's important to note that MTL network inference & loss computation, gradient computation, and model updates are fundamental and essential processes for any gradient-based optimization method.

As shown in Table 6, our analysis reveals that the time consumed by multi-agent learning and inference in TaskForce is significantly smaller in proportion to the time spent on the MTL network's task gradient computation, which typically constitutes the largest portion of gradient-based methods.

# F    STATISTICAL ANALYSIS OF EVALUATION RESULTS

Given that our methodology employs a Reinforcement Learning agent, discovering a suitable policy may fail. Consequently, we summarize the mean and variance of evaluation results across three trials for all experiments in the main manuscript in Table 7. Our approach, when relying solely on a loss-based reward, exhibits a non-negligible level of variance across trials. However, our method,

Table 6: Computational cost of TaskForce components on the NYU-v2 dataset.

| Procedure | Training cost (sample/ms) |
|---|---|
| MTL Network inference & compute loss | 62.02 |
| MTL Network gradient computation | 154.01 |
| MTL Network update | 4.78 |
| Agents inference & compute loss | 27.84 |
| Agents update | 17.04 |

incorporating the provably convergent gradient-based rewards $r_{\mathcal{G}}$ and $r_{\mathcal{G}}^{\text{IMTL}}$, demonstrates a comparatively low variance across the three trials. This suggests that gradient-based rewards can contribute to enhanced training stability.

Table 7: Summary of statistical analysis of evaluation results of main manuscripts. We report the 3-run mean and variance of our method under the different configurations.

| Method | NYU-v2 | Cityscapes | QM9 |
|---|---|---|---|
| STL (single-task learning) | 0.00 % | 0.00 % | 0.00 % |
| Ours ($r_{\mathcal{L}}$) | $-5.46\pm1.79$ % | $-0.33\pm0.06$ % | $+64.2\pm3.8$ % |
| Ours ($r_{\mathcal{L}}$ & $r_{\mathcal{G}}$) | $-6.47\pm0.51$ % | $-0.65\pm0.01$ % | $+59.0\pm0.8$ % |
| Ours ($r_{\mathcal{L}}$ & $r_{\mathcal{G}}^{\text{IMTL}}$) | $-6.23\pm0.83$ % | $-0.66\pm0.02$ % | $+61.5\pm1.7$ % |

Performing statistical analysis only on our method makes direct comparisons with other methods challenging. Therefore, we've summarized the results in Table 8 (including both reported and reproduced values with confidence intervals) for several existing methods and ours on the QM9 dataset below, as it represents the most challenging scenario with the highest number of tasks. Note that previous literature often tends to report higher values than what can be reproduced. For this reason, we used reported values for the main paper, and the statistical information for these reported values was sourced from the NashMTL (Navon et al., 2022) and Aligned-MTL (Senushkin et al., 2023).

Table 8: Statistical summary of evaluation results on the QM9 dataset. We present the 3-run mean and variance of our proposed method and baseline models, including metrics reported in their original papers and our own reproduced metrics.

| Method | $\Delta\mathbf{m}$ (reported) | $\Delta\mathbf{m}$ (reproduced) |
|---|---|---|
| STL (single-task learning) | 0.00 % | 0.00 % |
| LS | $+177.6\pm3.4$ % | $+177.1\pm3.5$ % |
| MGDA | $+120.5\pm2.0$ % | $+113.1\pm4.3$ % |
| IMTL | $+77.2\pm9.3$ % | $+77.9\pm4.1$ % |
| NashMTL | $+62.0\pm1.4$ % | $+63.1\pm1.6$ % |
| Aligned-MTL | N/A | $+81.9\pm2.3$ % |
| Ours ($r_{\mathcal{L}}$) | $+64.2\pm3.8$ % | $+64.2\pm3.8$ % |
| Ours ($r_{\mathcal{L}}$ & $r_{\mathcal{G}}$) | $+59.0\pm1.0$ % | $+59.0\pm1.0$ % |
| Ours ($r_{\mathcal{L}}$ & $r_{\mathcal{G}}^{\text{IMTL}}$) | $+61.5\pm1.7$ % | $+61.5\pm1.7$ % |

## F.1 ROBUSTNESS OF THE TASKFORCE W.R.T. HYPERPARAMETERS

TaskForce, which leverages multi-agent reinforcement learning, can exhibit varying performance depending on the agents' hyperparameters. Therefore, we conduct comprehensive experiments on the Cityscapes dataset, systematically varying the values of key components that can influence the agents' learning to observe the resulting performance changes. We focus on two primary components: (1) replay buffer length and (2) exploration noise scale, evaluating the impact of different values on the overall performance metric $\Delta\mathbf{m}$ (where lower is better). Note that, for agent exploration, we progressively decrease the scale of the Ornstein-Uhlenbeck (OU) noise (Lillicrap et al., 2015) applied to the policy during the exploration period, transitioning from an initial scale to a final scale.

Figure 2 illustrates the statistical analysis of evaluation results under different configurations across these key components. Firstly, as the replay buffer length increases, the sampling efficiency of stored transitions improves, leading to enhanced stability. These also result in both improved performance and reduced variance. Regarding the exploration noise scale, we observe that introducing appropriate exploration leads to performance gains compared to configurations without exploration (i.e., noise scale: [0.0, 0.0]). However, excessive exploration intensity results in decreased performance and increased variance, indicating instability.

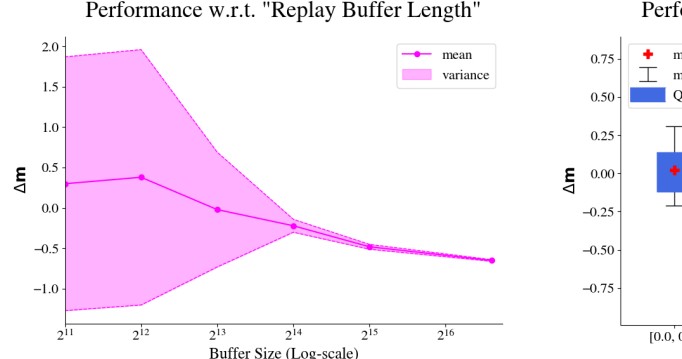

Figure 2: Performance trade-off of TaskForce on Cityscapes w.r.t. replay buffer size and exploration intensity. We report PSPNet (Zhao et al., 2017) model performance averaged over 3 random seeds.

## F.2 PRACTICAL SPEEDUP OF TASKFORCE

Our methodology employs highly compact agents (e.g., a 2-layer MLP with a hidden dimension of 64) to achieve a fast and practical Multi-Task Learning training process. However, the necessity to update the parameters of both agents and critics at each iteration poses an issue, as it still leads to a linear increase in training cost with respect to the number of tasks. We can mitigate this scalability issue by reducing the agent update frequency within TaskForce, which practically accelerates the Multi-task optimization process. Consequently, we investigate the performance trade-off resulting from variations in this update frequency.

We observe that reducing the agent update frequency leads to an increase in the optimization cost efficiency of TaskForce (a speedup of ×8.73 for an update frequency of 10). As shown in Figure 3, while increasing the update frequency tends to decrease performance, our method still outperforms existing approaches (e.g., Δm=0.01 for Aligned-MTL (Senushkin et al., 2023)), suggesting the potential for efficient optimization even in scenarios with a larger number of tasks.

## F.3 ALTERNATIVE FORMULATION OF THE TASKFORCE

As mentioned previously in our main manuscript, there are two strategies for interpreting the MTL training process as a cooperative Markov game:

*Strategy 1*: Viewing the entire MTL training process as a single episode (the methodology employed in the main manuscript).

*Strategy 2*: Alternatively, considering each gradient descent step as an individual episode (where transitions are derived using the same data point after a model update).

While *Strategy 1* treats data points as part of a non-stationary environment, *Strategy 2* allows for the measurement of the loss reduction rate on the same data point, potentially leading to more stable reward measurement. However, a significant drawback of *Strategy 2* is its requirement for nearly twice the training cost, as it necessitates measuring the loss and gradients of the network both before and after each update on the same data point. We provide the algorithmic description for this alternative formulation in Algorithm 2 and its experimental results on the Cityscapes dataset in Table 9. The approach utilizing Algorithm 2 enables the agent to learn more rapidly due to the stable

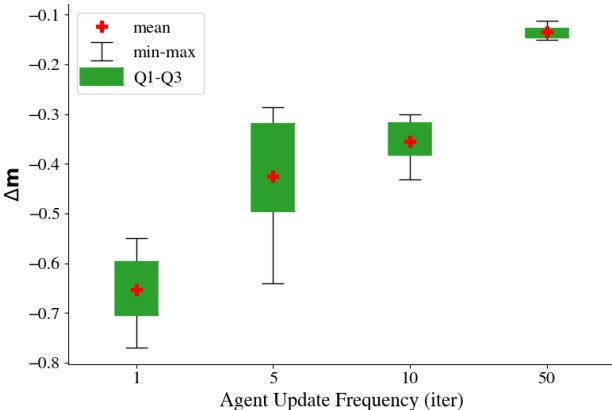

Figure 3: Performance trade-off of TaskForce on Cityscapes w.r.t. agent update frequency. We report PSPNet (Zhao et al., 2017) model performance averaged over 3 random seeds.

Table 9: Comparison between original TaskForce and Alternative formulation. We evaluate both methods on the Cityscapes 3-task setup. We report PSPNet Zhao et al. (2017) model performance averaged over 3 random seeds. We report the relative runtime of MTL training from the original TaskForce using the same computing resources.

| Method | Semseg. mIoU (%) ↑ | Instseg. L1 (px.) ↓ | Disparity MSE ↓ | $\Delta\mathbf{m}$ ↓ | runtime |
|---|---|---|---|---|---|
| STL | 66.73 | 10.55 | 0.33 | 0.00% | ×0.83 |
| TaskForce ($r_{\mathcal{L}}$) | 66.10 | 10.49 | 0.33 | −0.33% | ×1.00 |
| TaskForce ($r_{\mathcal{L}}$ & $r_{\mathcal{G}}$) | 66.63 | 10.55 | **0.32** | −0.65% | ×1.00 |
| TaskForce ($r_{\mathcal{L}}$ & $r_{\mathcal{G}}^{\mathrm{IMTL}}$) | 66.31 | **10.48** | 0.32 | −0.66% | ×1.00 |
| TaskForce-ALT ($r_{\mathcal{L}}$) | **66.81** | 10.51 | 0.33 | −0.41% | ×1.86 |
| TaskForce-ALT ($r_{\mathcal{L}}$ & $r_{\mathcal{G}}$) | 66.70 | 10.54 | **0.32** | −0.58% | ×1.86 |
| TaskForce-ALT ($r_{\mathcal{L}}$ & $r_{\mathcal{G}}^{\mathrm{IMTL}}$) | 66.50 | 10.49 | **0.32** | **−0.74%** | ×1.86 |

reward signal obtained from the same data point, resulting in a slight performance improvement of ($r_{\mathcal{L}}$) and ($r_{\mathcal{L}}$ & $r_{\mathcal{G}}^{\mathrm{IMTL}}$) settings. Nevertheless, given the marginal performance gain in Table 9 compared to the doubled computational cost, we adopt the first strategy in the main body of our work.

### F.4 VISUALIZE WEIGHT DYNAMICS

In Figure 4, we visualize the task weight dynamics observed during our experiments across the three datasets utilized: NYU-v2, Cityscapes, and QM9. Our method demonstrates the ability to adaptively determine appropriate weights for optimization across these datasets, including NYU-v2 and Cityscapes, as well as the relatively complex QM9 dataset, which presents a scale dominance problem and comprises a large number of tasks.

## G DISCUSSION: ADVANTAGES OF COOPERATIVE MULTI-AGENT RL IN MULTI-TASK OPTIMIZATION

In our main manuscript, we conduct ablation studies on key components of Cooperative MARL to evaluate each component's contribution. These results underscore the significant contribution of MARL's modular and cooperative architecture, especially when fully integrated, to improving both the stability of convergence and the overall performance in multi-task learning settings. Based on these results, this section aims to discuss the characteristics of cooperative MARL that may contribute to the observed performance improvements.

---

**Algorithm 2** Alternative Training Process of the TaskForce.

---

**Input:** data point number $N$, task number $T$, data points $\mathbf{X}, \{\mathbf{Y}_t\}_{1 \leq t \leq T}$, MTL model $\mathcal{F}(\cdot; \theta)$, agents $\{\mu_t(\cdot; \phi_t), Q_t^{\boldsymbol{\mu}}(\cdot, \cdot; \psi_t)\}_{1 \leq t \leq T}$, replay buffer $\mathbf{D}$, batch size of agents $b_{\text{agent}}$.

**Output:** trained MTL model $\mathcal{F}(\cdot; \theta^*)$, trained agents $\{\mu_t(\cdot; \phi_t^*), Q_t^{\boldsymbol{\mu}}(\cdot, \cdot; \psi_t^*)\}_{1 \leq t \leq T}$.

1: initialize $\mathcal{O}_{\text{prev}}, \mathcal{A}_{\text{prev}}, \mathcal{R}_{\text{prev}}$ as null matrix or 0.
2: **for** $i = 1$ **to** $N$ **do**
3:     sample data point $\{\mathbf{x}^i, \mathbf{y}_1^i, \cdots, \mathbf{y}_T^i\}$.
4:     # compute the current observation $\mathcal{O}$, and action $\mathcal{A}$.
5:     compute empirical loss set $\mathcal{L}(\theta)$ and task gradient set $\mathbf{g}$ from data point.
6:     generate observation $\mathcal{O}$ from $\mathcal{L}(\theta), \mathbf{g}$ by Equation 5.
7:     compute action $\mathcal{A} = \{\mu_1(o_1; \phi_1), \cdots, \mu_T(o_T; \phi_T)\}$ from $\mathcal{O}$.
8:     compute aggregated gradient $\mathbf{G}$ from $\mathbf{g}, \mathcal{A}$ by Equation 6.
9:     update MTL model $\mathcal{F}(\cdot|\theta)$ with $\mathbf{G}$ by $\theta \leftarrow \theta - \eta \mathbf{G}$.
10:     # compute the next observation $\mathcal{O}'$, and reward $\mathcal{R}$.
11:     compute next empirical loss set $\mathcal{L}_{\text{next}}(\theta)$ and next task gradient set $\mathbf{g}_{\text{next}}$ from same data point.
12:     generate next observation $\mathcal{O}_{\text{next}}$ from $\mathcal{L}_{\text{next}}(\theta), \mathbf{g}_{\text{next}}$ by Equation 5.
13:     compute reward $\mathcal{R}$ from $\mathcal{L}(\theta), \mathcal{L}_{\text{next}}$ by Equation 7-10.
14:     push transition $(\mathcal{O}, \mathcal{A}, \mathcal{R}, \mathcal{O}_{\text{next}})$ to replay buffer $\mathbf{D}$.
15:     **if** $i > b_{\text{agent}}$ **then**
16:         sample $b_{\text{agent}}$ transitions $\mathcal{T}$ from replay buffer $\mathbf{D}$.
17:         update actor $\mu_i(\cdot; \phi_i)$ and critic $Q_i^{\boldsymbol{\mu}}(\cdot, \cdot; \psi_i)$ with transitions $\mathcal{T}$ by Equation 11-13.
18:     **end if**
19: **end for**

---

**Problem Decomposition and Specialization** Unlike Single-agent RL, which attempts to learn a unified policy to handle all task dynamics, MARL assigns each task to a dedicated agent. This modular design enables agents to specialize in task-specific behaviors, potentially allowing the learning process to capture diverse optimization dynamics without interference. Such decomposition might be particularly beneficial in multi-task settings where task objectives may be partially conflicting.

**Improved Exploration through Distributed Policies** In MARL, agents can explore the optimization space in parallel, which not only enhances exploration coverage but also reduces the risk of premature convergence to suboptimal joint strategies. MARL leverages this by allowing task-specific agents to independently probe different gradient accumulation strategies, ultimately potentially leading to more effective joint optimization.

**Robustness and Redundancy** Multi-agent systems inherently offer robustness, as the failure or poor performance of one agent could be compensated by others. This redundancy leads to more stable learning dynamics, especially in noisy or partially observable environments. In contrast, a single-agent approach lacks the granularity and flexibility to adapt to individual task needs. It must implicitly learn to balance conflicting gradients and loss scales, often resulting in suboptimal convergence or instability in complex multi-task scenarios.

Taken together, our findings highlight the suitability of cooperative MARL, particularly MAD-DPG (Lowe et al., 2017), as a powerful optimization backbone for MTL frameworks. By decomposing task responsibilities and leveraging centralized training signals, MARL could offer both stability and efficiency, especially when facing task-level gradient conflicts and scale imbalances.

# H LIMITATION & FUTURE WORKS

Our method employs MADDPG, an early approach in Multi-Agent Reinforcement Learning (MARL). Consequently, the robustness of our optimization framework with more advanced MARL techniques remains unverified, as the optimization outcome can vary depending on the agents' performance. We believe that future research could further enhance multi-task optimization performance by applying more sophisticated MARL methodologies, a direction we leave for future investigators. Furthermore, while designing TaskForce, we consider a simultaneous training scenario and thus engineer very compact agents in terms of size and observation space. Nevertheless, a scalability issue persists, as the number of required agents still increases linearly with the number of tasks.

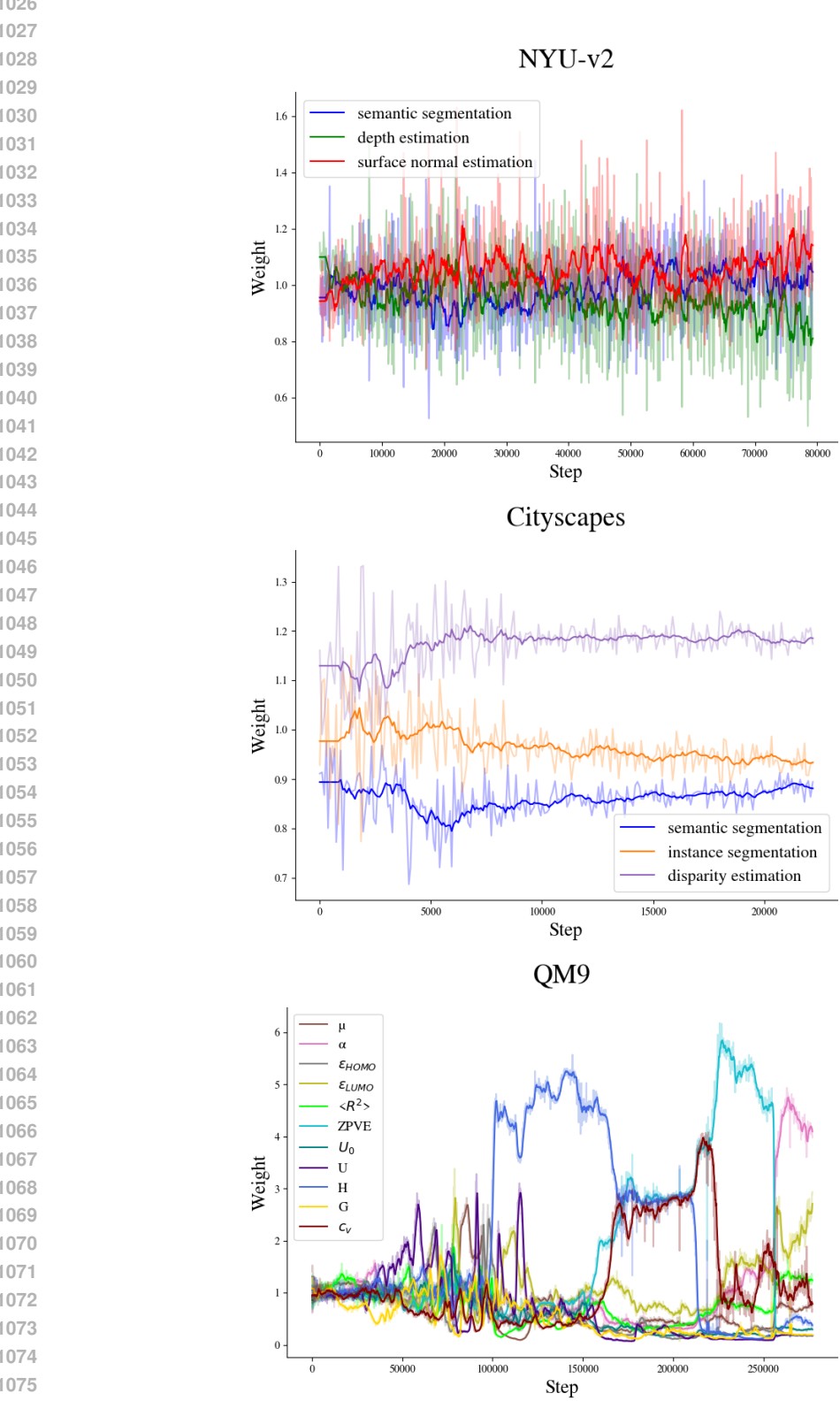

Figure 4: Task weight dynamics of NYU-v2, Cityscapes, and QM9 datasets. For improved visual-ization, we smooth the weight dynamics by using a moving average with a window size of 10.

## I  ETHICS STATEMENT

Following ICLR 2026 guidelines, we disclose that a Large Language Model (LLM) was utilized for assistance with grammar correction and text polishing. All research contributions, experimental results, and scientific claims are entirely the work and responsibility of the authors.

