# OpenReview forum: "TaskForce: Cooperative Multi-agent Reinforcement Learning for Multi-task Optimization"
_ICLR.cc/2026/Conference — ICLR 2026 Conference Withdrawn Submission_

### Official Review · Reviewer_b8s2 · 2025-10-25

**Soundness:** 4
**Presentation:** 4
**Contribution:** 3
**Rating:** 4
**Confidence:** 3

**Summary:**

This paper proposes a new method to address the open challenge of training in multi-objective learning, a topic that has gained considerable attention in recent years. The authors introduce a cooperative multi-agent reinforcement learning (MARL) framework, where agents learn to develop an effective joint optimization strategy. Each agent predicts balancing parameters that determine the weight of its contribution to the final gradient update. The proposed algorithm is evaluated against multi-task learning (MTL) baselines on the NYU-v2, Cityscapes, and QM9 datasets.

**Strengths:**

The paper is overall well-written and well-motivated within the context of existing literature.
The proposed method is innovative and relevant to the field.
The approach is conceptually sound and could have a meaningful impact on the multi-objective learning community.

This paper presents an interesting and original idea that connects cooperative multi-agent reinforcement learning with multi-task optimization. Conceptually, the approach is promising and could inspire further research in this direction.

**Weaknesses:**

The paper lacks theoretical guarantees or even an intuitive justification for the proposed framework. For instance, a conceptual figure similar to Figure 1 in previous works (e.g., Aligned-MTL or NashMTL) could help illustrate the underlying intuition.
The experimental results are not sufficiently convincing:
(a) The range of experiments is limited compared to prior works such as Aligned-MTL (Table 3) and NashMTL (10 seeds in Table 4).
(b) In Table 1, results are reported with only two significant digits, whereas NashMTL reports four. This discrepancy can meaningfully affect the interpretation of Δm.
(c) No results are provided for the 11 tasks of the QM9 dataset.
(d) The paper does not include comparisons of runtime or computational efficiency with previous algorithms.
(e) Overall, the reported improvements do not appear significant enough to support the claim that their algorithm achieves consistent improvements.

The current version lacks sufficient theoretical grounding and empirical rigor to fully support its claims. Strengthening the theoretical motivation, expanding the experimental analysis, and improving the presentation of results would substantially enhance the paper’s impact and credibility.

**Questions:**

Please address the weaknesses.

---

### Official Review · Reviewer_5YFF · 2025-10-30

**Soundness:** 2
**Presentation:** 3
**Contribution:** 2
**Rating:** 4
**Confidence:** 4

**Summary:**

This paper explores the problem of multi-task learning (MTL) through the lens of reinforcement learning (RL). The authors employ an actor–critic framework to dynamically predict task weights during training. Specifically, the observation space consists of the concatenation of the Gram matrix and task losses, while the action corresponds to the predicted task weights. The reward is designed to encourage minimization of all task losses, thereby guiding the learning policy toward balanced optimization across tasks. Extensive experiments are conducted to demonstrate the effectiveness of the proposed method.

**Strengths:**

1. Studying MTL from the perspective of RL is novel and provides a fresh direction for adaptive task weighting.

2. The paper is generally well-organized.

3. The proposed method achieves competitive performance under certain experimental settings, demonstrating its potential effectiveness.

**Weaknesses:**

1. The proposed RL-based paradigm may face scalability issues in many-task scenarios, as the number of agents grows linearly with the number of tasks, making it difficult to generalize efficiently.

2. The actor-critic framework introduces additional trainable parameters for task weight prediction, which could lead to an unfair comparison with conventional MTL baselines. It would be helpful to investigate whether employing a simpler neural network to predict task weights could achieve comparable results.

3. The experimental comparison is somewhat limited. Including more recent and competitive MTL approaches[1][2] would strengthen the evaluation.

4. Additional evaluations on more diverse benchmarks (e.g., CelebA) are expected to better demonstrate generalization.

5. The paper lacks theoretical insights or analysis to justify the effectiveness and convergence of the proposed RL-based optimization.

6. The claim that “existing methods either lack the stochasticity needed to escape poor local minima or fail to explicitly address conflicts at the gradient level” is not sufficiently supported.

Reference:

[1] Fair resource allocation in multi-task learning. ICML 2024.

[2] Towards consistent multi-task learning: Unlocking the potential of task-specific parameters. CVPR 2025.

**Questions:**

Please refer to the Weaknesses.

---

### Official Review · Reviewer_KK4j · 2025-11-01

**Soundness:** 3
**Presentation:** 3
**Contribution:** 3
**Rating:** 6
**Confidence:** 3

**Summary:**

TaskForce introduces a novel framework for multi-task learning (MTL) that formulates the multi-task optimization (MTO) problem as a cooperative multi-agent reinforcement learning (MARL) game. Each task is modeled as an agent that learns a policy to adaptively balance multiple tasks based on training feedback, instead of relying on heuristic rules. The method leverages the Gram matrix of task gradients to reduce dimensionality while retaining information about gradient magnitudes and inter-task alignments. A hybrid reward function combines gradient- and loss-based optimization signals, and each agent outputs a weight for its task’s gradient. The proposed approach is evaluated on three standard benchmarks. Overall the manuscript presents an interesting idea and a step away from fixed heuristics for MTL balancing and avoidance of interference to a learnable policy. However, theoretical grounding is thin, and I have some questions regarding the experiments and baselines.

**Strengths:**

Strengths

The paper proposes a distinctive formulation that sets it apart from the large body of heuristic-based task-weighting and gradient-aggregation methods.

The method is clearly motivated and well-documented, and several ablation studies are included (see Appendix A).

The use of a cooperative MARL framework is creative and potentially extensible to broader optimization settings.

**Weaknesses:**

Weaknesses

The paper repeatedly frames r_G as “grounded in provably convergent criteria', but there is no convergence guarantee for the learned stochastic policy with mixed rewards and non-stationary dynamics. Otherwise the theoretical grounding is fairly thin to make this a fundamental progress.

The proposed method introduces many more degrees of freedom and higher computational overhead for relatively limited empirical gains.

The theoretical link between cooperative MARL and general MTL optimization remains somewhat underdeveloped.

The precise set-up of the baselines in function of gradient normalization is unclear for the QM9 benchmark.

I detail the latter three points more extensively in the "questions' section.

**Questions:**

Detailed Comments and Questions

Can you show that r_g + r_l avoids failure modes where pure ∥∑wg∥ minimization collapses?

Information loss in Gram matrix compression: The Gram matrix discards directional information beyond pairwise inner products, meaning distinct gradient constellations can share the same Gram matrix. Have you observed failure modes due to this compression—for instance, in degenerate or near-collinear task settings?

Sensitivity to λ_G: Performance appears to hinge on a very small λ_G. How sensitive are results to this hyperparameter, and what happens if λ_G is increased by an order of magnitude?

Table 3 inconsistencies: Table 3 mixes relative normalizations, includes a “rough calculation” for the non-Gram variant, and lists two rows with identical “×1.00” costs despite different configurations, while one DE variant shows “×3.21” contradicting claims that DE improves efficiency. Please clarify this with consistent baselines and real wall-clock measurements, or move these results to an appendix with clearer methodology.

QM9 underperformance vs. STL: All MTL methods—including TaskForce—perform worse than single-task training on QM9. Could this stem from architectural mismatch or excessive parameter sharing in relatively low-dimensional task heads? Clarifying this would help interpret the results. (A public code link would also be appreciated for verification.)

Gradient normalization fairness: You mention applying gradient normalization as part of TaskForce’s setup, but it’s unclear whether the same normalization was applied to baselines. Since TaskForce agents operate on normalized Gram matrices and losses (after normalization), ensuring fair comparison is important—especially on QM9, where performance differences are partly attributed to task-scale imbalances.

Choice of MARL technique: How sensitive is performance to the selection of MARL algorithm (e.g., MADDPG vs. alternatives) or to the update frequency of the agents? A brief discussion would help contextualize robustness.

Interpretation of weight dynamics (Fig. 4): Figure 4 shows evolving task weights, but no analysis is given. Can you interpret these dynamics—e.g., are they correlated with Gram-matrix entries, loss trends, or signs of conflict resolution among tasks?

**Details Of Ethics Concerns:**

No concerns

---

### Official Review · Reviewer_inyc · 2025-11-01

**Soundness:** 3
**Presentation:** 3
**Contribution:** 2
**Rating:** 4
**Confidence:** 3

**Summary:**

This paper introduces TaskForce, a novel framework that reformulates multi-task optimization as a cooperative Multi-Agent Reinforcement Learning (MARL) problem.
TaskForce models each task as an agent that observes compact summaries of task gradients and losses and outputs a weight determining its contribution to the shared update. And the hybrid reward function combining loss reduction and gradient alignment further boost the performance on NYU-v2, Cityscapes, and QM9 datasets, achieving the SOTA results.

**Strengths:**

- The paper is well organized with a good writing.
 - It is novel to solve multi-task optimization task in Multi-Agent Reinforcement Learning way.
 - The use of centralized critic and shared cooperative reward stabilizes multi-agent learning and provides consistent convergence
 - The evaluation across diverse benchmarks achieves SOTA and shows robustness

**Weaknesses:**

- Benchmarks focus on homogeneous visual or molecular tasks. Cross-modality tasks remain untested.
 - Implementing MADDPG with multiple agents and critics increases hyperparameter sensitivity and tuning cost, there are concerns about **reproducibility**.
 - Learned weight policies lack interpretability; it is unclear which task dominates or how cooperation evolves during training.

**Questions:**

1. How are the two weights in the reward function determined? Are there any adaptive methods?
2. Nowadays, MLLM also face many multi-task scenarios, so why are experiments still only conducted on traditional models?

---

### Note · Authors · 2025-11-12

I have read and agree with the venue's withdrawal policy on behalf of myself and my co-authors.